# Emissions Reduction Strategies for the Orange and Cherry Industries in New South Wales

**DOI:** 10.3390/foods12183328

**Published:** 2023-09-05

**Authors:** Aaron T. Simmons, Marja Simpson, Paul-Antoine Bontinck, John Golding, Tim Grant, Jess Fearnley, Steven Falivene

**Affiliations:** 1NSW Department of Primary Industries, Muldoon St., Taree, NSW 2430, Australia; 2School of Business, University of New England, Elm Ave, Armidale, NSW 2350, Australia; 3NSW Department of Primary Industries, Orange Agricultural Institute, Orange, NSW 2800, Australia; 4Lifecycles, 2/398 Smith Street, Collingwood, VIC 3066, Australia; paul-antoine@lifecycles.com.au (P.-A.B.);; 5NSW Department of Primary Industries, Locked Bag 26, Gosford, NSW 2250, Australia; john.golding@dpi.nsw.gov.au; 6NSW Department of Primary Industries, P.O. Box 62, Dareton, NSW 2717, Australia

**Keywords:** pyrolysis, green nitrogen, revegetation, carbon neutral, photovoltaic

## Abstract

The orange and cherry industries in New South Wales, Australia, are major horticulture industries with a high export value. Climate change has resulted in the carbon footprint of products being used by consumers to guide purchases meaning that products with a relatively high carbon footprint risk losing market access. The carbon footprint of cherry and orange production is unknown and there is no assessment of the success of climate change mitigation strategies to reduce the carbon footprint of their production and move production towards being carbon neutral. This study assesses the climate change mitigation potential of five management changes to on-farm cherry and orange production (revegetation, the use of nitrification inhibitors, renewable energy, green N fertilisers, and pyrolysis of orchard residues) over a 25-year period. for example, orchards in relevant growing regions. The results show that the carbon footprint of production can be reduced by 73 and 83% for cherries and oranges, respectively, when strategies that avoid emissions are included in their production. When strategies that sequester C from the atmosphere are also included, cherry and orange production becomes C negative in the first few years of the scenario. The economics of implementing these strategies are unfavourable, at present; however, our results indicate that the NSW cherry and orange industries can be confident in achieving emissions reductions in on-farm production to assure market access for their products.

## 1. Introduction

Citrus and cherry production are important horticultural industries for Australia and the state of New South Wales (NSW), Australia. Globally, in 2021, Australia was the 25th largest producer of oranges with an annual production of 456,719 tonnes compared to the largest producer, Brazil, with an annual production of 16,214,982 tonnes [1]. NSW was the largest orange-producing state in Australia with the majority grown in the Riverina region. In 2021–2022, 260,557 tonnes of oranges were produced in NSW with a significant proportion being exported to Asia as the primary destination with a gross value of AUD 39.5 m [2]. There are two main varieties grown in NSW, Navel and Valencia, with a smaller area being planted to Valencia. Australian cherries are exported to more than 30 countries. In 2021, Australia was the 19th largest cherry producer globally, with an annual production of 21,309.94 tonnes. Türkiye was the largest producer with an annual production of 689,834 tonnes (FAO 2023). NSW was a major contributor to Australia’s cherry industry, producing more than 4812 tonnes in 2021–2022 with a gross value of AUD 62.6 m with the main growing regions located in Orange and Young [2]. Cherries are considered a high value crop and NSW exported 950 tonnes in 2022 and 612 tonnes of cherries in 2021 with Asia being a major market [2]. For horticulture, as for any agricultural system, greenhouse gas (GHG) emissions are emitted during production and are a concern for the industry and consumers. Sources of direct and indirect GHG emissions in the perennial horticulture industry include the production and use of liquid fuels (e.g., diesel) and electricity, the production and use of nitrogenous fertilisers, and plant-protection products. Studies examining the GHG emissions associated with perennial horticultural crops are limited; however, one study assessed the 23 most commonly grown annual horticultural crops in Australia [3]. That study showed that 65% of total GHG emissions were emitted from energy generation for irrigation and processing, 17% from on-farm emissions associated with fertiliser use, 10% from agro-chemicals, 7% from fossil fuel use on the farm, and 1% from on-farm machinery. Although N_2_O emissions associated with horticultural production are relatively low, these emissions are expected to increase due to an increase in fertiliser use to meet global food and fibre demands [4].

The concern by consumers has led to a consumer preference for low-GHG-emission produce [5], and the introduction of initiatives for products to declare their carbon footprint (e.g., the European Commission’s Product Environmental Footprint) will facilitate consumers in choosing low-GHG-emissions foods. At present, there is no requirement for Australian horticulturists to undertake carbon accounting for their businesses; however, as the market demand for low-GHG-emissions foods increases, growers may need to provide a carbon footprint for their produce and demonstrate an ongoing reduction in their carbon footprints to maintain market access.

There are, at present, opportunities to reduce the GHG emissions intensity of horticultural systems. For example, nitrification inhibiters (NIs), urease inhibitors, and biochar can reduce N_2_O emissions and improve the yield [6,7], and have been included as a mitigation strategy for agricultural N_2_O emissions by the Intergovernmental Panel on Climate Change (IPCC) [8]. Amongst NIs, 3,4-dimethylpyrazole phosphate (DMPP) is widely used and has been shown to be effective in reducing NO_3_ leaching and N_2_O emissions when granular nitrogen (N) fertilisers are coated or DMPP is included in fertigation systems [9]. The research demonstrates that using DMPP can reduce N_2_O emissions associated with inorganic fertiliser use by up to 100% [10]; although, other research reported lower reductions [11,12] or no reductions at all [13]. Using DMPP can increase the nitrogen use efficiency (NUE) in irrigated production systems, and the improvement of NUE is dependent on the N application rate [14]. Adding biochar, derived from pyrolysis (i.e., thermal degradation of biomass in an oxygen-limited environment) to soils can also reduce N_2_O emissions associated with the use of fertilisers; however, this is unlikely to reduce it to the same extent as the use of NIs [6].

Biochar itself is also a recognized climate change mitigation strategy [8] because it takes C that was sequestered in vegetation from the atmosphere and stabilises it. Pyrolysis is recognised by the IPCC as a carbon dioxide removal (CDR) strategy [15]. Vegetation is a common source of biomass for pyrolysis, and orchards generate biomass from mature trees that are removed during replanting and from on-going maintenance, such as pruning.

Revegetation is also a climate change mitigation strategy because trees sequester carbon from the atmosphere and, although revegetation of parts of a farm may reduce the area that is planted with trees, it may also offer co-benefits, such as windbreaks to decrease fruit damage and provide a habitat for natural substances to improve biological pest control outcomes [16].

GHG emissions associated with the production and combustion of fossil fuels for orchard operations have the potential to be mitigated by electrifying farm machinery when electricity is generated from renewable sources [17]. Using renewable electricity to power pumps for irrigating orchards can also reduce the emissions associated with horticultural production [18].

Pathways for emissions reductions in the NSW horticulture sector to ensure the demand for low-emissions products in international markets are yet to be assessed. The purpose of this study is to evaluate five GHG mitigation strategies for two key horticultural crops in NSW, cherries and oranges, and to identify the potential pathways to carbon neutrality for the perennial horticulture sector more broadly.

## 2. Materials and Methods

### 2.1. Boundaries

A system boundary of cradle-to-farm-gate was used. This meant that the emissions associated with the production, transport, and use of all inputs, such as fertiliser, plant protection products, and orchard operations, were included in the assessment. A temporal boundary of 20 years was used to account for annual increases in carbon sequestered in vegetation, as described in Section Revegetation.

### 2.2. System Descriptions

#### 2.2.1. Lapin Cherries

An irrigated cherry orchard cv. Lapin in the Orange region of NSW with a size of 20 ha was used for the study. Cherry orchards in the regions of study are redeveloped when trees reach an age of approximately 20 years old; therefore, it was assumed that 5% of the orchard was redeveloped on an annual basis. The orchard was assumed to have a density of 1000 trees ha^−1^.

#### 2.2.2. Navel Oranges

An irrigated orange orchard cv. Washington Navel in the Sunraysia region of NSW with a size of 20 ha was used. The orchard was assumed to have a density of 519 trees ha^−1^. Orange orchards in the regions of study are redeveloped when trees reach an age of approximately 20 years old; therefore, it was assumed that 5% of the orchard was redeveloped on an annual basis.

### 2.3. Climate Change Mitigation Strategies

Climate change mitigation strategies can be categorised into two groups: avoided emissions or carbon sequestration. Avoided emissions are strategies that reduce the GHGs emitted relative to current practices and are not subject to reversals. Carbon sequestration strategies remove carbon from the atmosphere and retain carbon either temporarily or permanently, depending on the strategy, and some strategies can also be subject to reversals.

#### 2.3.1. Avoided Emissions

##### Nitrification Inhibition

The use of the nitrification inhibitor 3,4-dimethylpyrazone phosphate (DMPP) when applying inorganic nitrogenous fertilisers via fertigation to orchards was assumed to reduce N_2_O emissions associated with fertiliser use. The research demonstrates that the use of DMPP can reduce N_2_O emissions in the citrus and cherry orchards by 55% based on a recent meta-analysis [19]. The use of nitrification inhibitors can also increase the efficiency of N fertiliser [14] that can reduce leaching and the atmospheric deposition of N from volatilised fertiliser; however, the impacts of these effects were not included in this study due to limited evidence.

##### Renewable Energy

Using renewably generated electricity to power tractors was also used as an emissions reduction strategy. Emissions associated with tractor production assumed the tractor had a 211 kW motor with a 62 kWh battery. Two batteries were used for the tractor so that one could be charged while the other was in use. A 20 kW solar array was used to ensure the batteries could be recharged in one day and allow for the consistent daily use of the tractor. Emissions associated with the production of the tractor and photovoltaic panels were amortised over the lifetime of the tractor (assumed to be 7000 h). The electric tractor was only used for maintenance activities (e.g., spraying and mowing) in the orchard and diesel-powered tractors that we required were retained for more intensive activities, such as tree stump pulling.

Replacing grid-sourced electricity with renewable electricity for pumping irrigation water was also used as an emissions reduction strategy. For cherry production, it was assumed that 154 W was required to move 1000 L of water [20]; therefore, it was assumed that a 20 kW solar array had the capacity to pump 0.13 ML of water per hour. For orange production, it took 309 W to move 1000 L of water from the pump; therefore, a 40 kW solar array power was assumed to provide the capacity to pump 0.13 ML of water per hour. The emissions associated with the production of solar panels to power the pump was amortised over the lifetime of the pump that was assumed to be 15 years.

##### Green N

The generation of N for use in fertilisers is GHG intensive [21]; therefore, an assumption that nitrates, in the form of nitric acid, were generated using a low-GHG-emissions process (e.g., the proprietary technology of Nitricity) was included to estimate reductions in scope 3 emissions from using green N.

#### 2.3.2. Carbon Sequestration

##### Pyrolysis

Pyrolysis, the heating of organic materials in the absence of oxygen can stabilise organic carbon in the form of charcoal. The 5% of trees that was removed each year for redevelopment was assumed to be chipped and pyrolysed. It was assumed that 30% of pyrolysed dry matter was recovered as biochar with a carbon content of 80% (S. Joseph, pers. comm.). The remaining carbon in the biomass was assumed to be emitted as CO_2_ after syngas created by the pyrolysis process was combusted converting CH_4_ to CO_2._ It was assumed that biochar was applied to soils where it would slowly degrade releasing CO_2_ back into the atmosphere [22], and this was included as an emission. We assumed that high-temperature pyrolysis was used and, because we were uncertain where biochar would be applied to the soils, a soil temperature of 15 °C was assumed, resulting in 18% of carbon stored in the biochar being released back into the atmosphere over a 100-year period [22]. The loss function that describes this was linear; therefore, we assumed that 0.18% of carbon in the biochar was released back into the atmosphere annually.

To estimate the C in tree biomass that was available for stabilisation in the biochar each year, we used published values. For the cherry orchard, the mass of a tree on a dry matter basis was assumed to be 82 kg tree^−1^ [23] based on the values for 20-year-old *Prunus cerasus* trees planted at a rate of 292 trees ha^−1^. Multiplying the mass of a tree by the planting density assumed in the present study, and assuming a 50% carbon content of biomass, produced a carbon mass of 41,000 kg ha^−1^, or 2050 kg carbon available for stabilisation in pyrolysis when 5% of the orchard was removed per annum. For the orange orchard, a carbon mass of 102 kg tree^−1^ was assumed [24]. This was based on the values provided for 16-year-old trees of *Citrus sinensis* cv. Tarocco Sciré planted at a rate of 494 trees ha^−1^. Multiplying the carbon mass of a tree by the planting density produced a carbon mass of 52,224 kg carbon ha^−1^ with 5% (2611 kg) of carbon available for stabilisation in pyrolysis per annum due to the rate of re-planting.

##### Revegetation

Revegetation was used to strategically revegetate areas of the farm to native trees to sequester atmospheric carbon, and where revegetation occurred, it was assumed to displace orchard production. The calculation of the area that was planted to shelterbelts was performed assuming that the 20 ha orchard had the dimensions of 400 × 500 m with a 15 m shelterbelt on the northern, western, and southern boundaries, and one shelterbelt through the centre of the orchard on an east–west axis. This resulted in 3.45 ha (or 17%) of the orchard being dedicated to shelterbelts. The planting of shelterbelts was assumed to occur when 1 ha was redeveloped; therefore, the area of the orchard that was dedicated to shelterbelts increased linearly over time from 1% when the first block was developed to the maximum of 17% when the final block was redeveloped in the 20-year period.

The carbon sequestered via revegetation was estimated using the Australian government Full Carbon Accounting Model (FullCAM) as set out in the methods for estimating sequestration under the Environmental Plantings method [25] for the emissions reduction fund parameterised with GPS co-ordinates of Balranald, NSW, and Orange, NSW, for the Riverina and Orange regions, respectively [25].

### 2.4. Data Sources

The data for on-farm operations, inputs, and yield of a Valencia orange orchard were obtained from NSW DPI gross margins of production [26], as were the operations and inputs for a cherry orchard [27]. These gross margins were developed by regional experts in the relevant production system and represent best-practice management. The data assumptions for yields, inputs, and orchard operations are presented in the Appendix A.

### 2.5. GHG Emissions and Sequestration Calculations

GHG emissions associated with the use of nitrogenous fertilisers and the use of lime were calculated according the Australian National Greenhouse Gas Inventory [28], and GHG emissions associated with the combustion of liquid fuels on-farm were calculated using the appropriate emissions factor from the Australian National Greenhouse Accounts [29]. Pre-farm GHG emissions (i.e., emissions associated with the production and transport of inputs, such as fertilisers and plant protection products) were obtained from life cycle inventory databases, ecoinvent v3.9 [30], and AusLCI [20]. The ecoinvent database is the largest transparent unit-process life cycle inventory database globally available and he AusLCI is a scientifically robust, standardised, and transparent database specific to Australian agriculture production.

Though we assumed that 5% of the orchard was redeveloped each year, the carbon sequestered in the fruit trees was not included in our calculations. This was because when the carbon stocks were assessed across the entire orchard, there was no annual change as was demonstrated for forestry estates that were rotationally harvested [31].

The global warming potential values for a time horizon of values for 100 years (GWP_100_) for CH_4_ from fossil sources, CH_4_ from natural sources, and N_2_O were collected from the IPCC AR5 report and were 30, 28, and 265, respectively [32].

The emissions intensity of production was calculated by dividing the total yield of the orchard by the sum of GHG emissions and sequestration.

## 3. Results

### 3.1. Yield

A 1 ha block of the orchard was redeveloped each year and shelterbelts were added each time a 1 ha block was redeveloped. This resulted in a reduction in the area of land dedicated to fruit production and resulted in the total productivity of cherry and orange orchards declining over time. At the end of the 20-year period, the total productivity of the cherry declined from 245 t to 203 t annum^−1^ and the total productivity of the orange orchard declined from 501 to 416 t annum^−1^ (Figure 1).

### 3.2. Emissions Intensity

The emissions intensities of producing 1 kg of cherries and oranges using current production methods, as described above, were 137 and 128 g CO_2_-e, respectively. The greatest contributor to the emissions intensity of cherries was the emissions associated with the production of fertiliser, and for oranges the greatest contributor to GHG emissions intensity was electricity associated with irrigation (Table 1).

### 3.3. Emissions Reductions Strategies

Applying emissions reductions strategies of nitrification inhibitors, renewable energy, and green N reduced the emissions intensities gf cherry and orange production by 75 and 83%, respectively. The greatest emissions reductions, expressed as a % of emissions under current production, occurred when pumps for irrigation were moved from grid-powered to solar-powered electricity (Table 1). For both crops, emissions reductions associated with replacing nitrates with ‘green’ nitrates and the use of a nitrification inhibitor to reduce emissions associated with fertiliser use and replacing diesel-fuelled tractors with electric tractors also made considerable contributions to emissions reductions outcomes (Table 1).

### 3.4. Carbon Sequestration

The two carbon sequestration methods included in this study further reduced the emissions intensity of producing 1 kg of cherries and oranges. When carbon sequestered in shelterbelts was included, the emissions intensities in year 1 of the 20-year scenario were 79 and 86% lower than the emissions intensity under current production for cherries and oranges, respectively. Furthermore, the production of cherries became carbon neutral in year 16 and the production of oranges became carbon neutral from year 10 onwards (Figure 2). When the carbon sequestered in the vegetation and carbon sequestered in biochar were combined with the emissions reductions strategies, the emissions intensity of cherry production was reduced by 99% and became carbon neutral in the 2nd year of implementation, while orange production became carbon neutral in year 1 with a 113% reduction in GHG emissions relative to current production (Figure 2).

### 3.5. Total Emissions

The cumulative total GHG emissions for each orchard over the 20-year period was calculated (Figure 3). Where current management was assumed for the 20-year period, the total GHG emissions were 1372 and 1278 t CO_2_-e for cherry and orange production, respectively. Where all emissions reduction strategies were implemented and carbon sequestration in trees was included, the cherry orchard was responsible for emitting 1273 less t CO_2_-e than current management, and when carbon sequestration in vegetation and biochar were included, the orchard was responsible for 1536 less t CO_2_-e. For orange production, where emissions reductions and carbon sequestered in trees were included, 1290 less t CO_2_-e was emitted, and when carbon sequestered in biochar was added, 1617 less t CO_2_-e was emitted.

## 4. Discussion

The results of this study suggest that it is possible for cherry and orange orchards in NSW to implement changes to their production systems so that they positively contribute to the climate change problem by being a carbon sink. A mix of different emissions reduction strategies can be utilised; however, our analysis suggests that the pyrolysis of trees, when the redevelopment of the orchard occurs, is essential for carbon neutrality to be achieved (Figure 3). This is because carbon sequestered from the atmosphere and stored in trees is stabilised as biochar when it is pyrolysed, thereby removing and storing carbon from the atmosphere in a long-term capacity. The pyrolysis of plant materials is a recognised carbon dioxide-removal strategy [15]; however, this research is the first to demonstrate the importance of this strategy in horticultural production achieving carbon neutrality. No research has identified the pathways to carbon neutrality for cherry production; however, one study assessed the pathways to carbon neutrality for orange production [33]. That study demonstrated that Chinese citrus production could become a carbon sink by optimising nitrogen fertiliser use, replacing 50% of chemical fertilisers with organic fertilisers and the use of cover crops as green manures to supply N. Those strategies are not relevant to the systems we assessed here because N fertiliser is not overapplied, as occurs in the Chinese systems; the supply of organic fertilisers to replace chemical fertilisers are constrained and the environment in which oranges are grown is not conducive to growing green manure crops. Nevertheless, both that study and the present study recognise that a more sustainable production of N is required for carbon neutrality to be achieved.

This study assessed the potential for five strategies to reduce the GHG emissions associated with cherry and orange production in NSW. However, the scope for greater emissions reductions exist but were not included in the present study due to a lack of data and/or a lack of technological readiness. For example, the emissions from diesel production and combustion were reduced by between 48 and 77%, depending on crop type; however, these emissions can be reduced further if high-horsepower machinery required for the redevelopment of orchards is electrified. There is also the scope to increase carbon sequestration by pyrolysing orchard prunings; however, the mass of prunings from an orchard on an annual basis cannot be quantified for this study. Potential also exists to cut back shelterbelts when orchards are redeveloped and the carbon stored in shelterbelts is pyrolysed. We assumed that nitrates were sourced from green production systems; however, systems to produce ammonia, another form of N used in fertilisers, are being developed and would further reduce the emissions associated with the production of inputs, such as urea (e.g., Jupiter Ionics). Emissions associated with fertiliser use can be reduced further where the use of NI improves NUE [11,12] allowing the mass of N fertiliser applied to crops to be reduced. Finally, maximising the use of integrated disease and pest management has the potential to reduce the emissions associated with the production and application of plant protection products.

The pathways to carbon neutrality assessed here are relatively mature with the technological readiness of four of the five climate change mitigation strategies assessed here considered high (i.e., technological readiness level (TRL) of 7–9). For example, nitrification inhibitors, such DMPP, are commercially available for producers to use, photovoltaic technologies are widely used globally to generate electricity, the planting of trees to sequester atmospheric carbon is a common method to generate carbon credits [25], and methods also exist to generate carbon credits by pyrolyzing organic materials [34]. The remaining strategy of using ‘green’ nitrates is of a lower TRL and is not commercially available at present; however, companies with proprietary technology (e.g., Nitricity) are scaling up production to meet the projected global demands for green N. Electric tractors are close to being produced [35] and companies that specialise in the retrofitting of electric motors to diesel tractors exist (e.g., Janus technologies).

The economics of moving citrus and cherry production to achieve carbon neutrality is uncertain and we recognise that it is unlikely to be cost effective at present; however, the imperative to reduce GHG emissions is likely to increase as consumers choose lower-emissions products [5] and global initiatives (e.g., the European commission product environmental footprint [36]) roll out to support consumers in making more environmentally friendly choices. The Australian horticultural sector does not have a target for carbon neutrality at present, unlike other agricultural sectors that are under greater pressure from markets due to their relatively high emissions intensity (e.g., the red meat sector that has set a carbon neutrality target of 2030 [37]). The results from this study indicate that the horticulture industry can set an ambitious target of carbon neutrality with pathways to support the process. Including the economic feasibility of the emissions reduction pathways was out of scope in this exploratory study. Therefore, further research should investigate the economics of the mitigation pathways demonstrated in this study and the potential trade-offs that may need to be considered. For example, the cost of utilising NI may be offset to some extent by the need to apply less nitrogen. Investments in solar for water pumping is a significant initial capital cost but would reduce the operating costs for a business. As more businesses begin implementing emissions reductions strategies and compete for customers, the economics of investing in technologies that support the move, carbon neutrality may become more favourable.

## 5. Conclusions

The carbon neutrality of cherry and orange production in NSW is technically possible; however, the economics of implementing strategies to achieve this is uncertain and should be investigated in further studies. As governments are implementing policies for emission reduction targets, incentive schemes can increase the uptake of emission reduction strategies. The strategies assessed here are also applicable to other perennial horticultural crops and warrant further studies to assess the technical potential for other crops to achieve carbon neutrality. Such studies can inform the Australian horticultural industry in setting on-farm GHG emissions reductions targets to support ongoing social licence for production to ensure market access is maintained.

## Figures and Tables

**Figure 1 foods-12-03328-f001:**
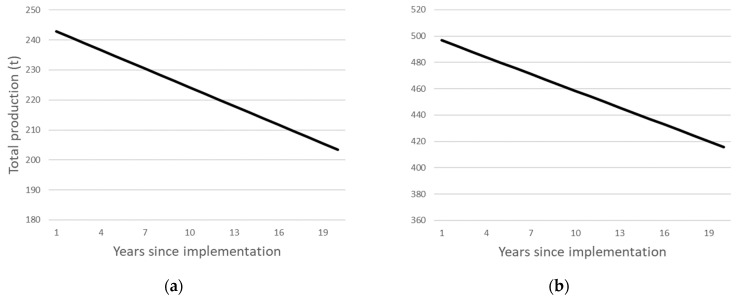
Total production (t) over a 20-year period for (**a**) cherry cv. ‘Lapin’ and (**b**) orange cv. ‘Washington’ orchards of 20 ha when shelterbelts to sequester atmospheric CO_2_ that reduce the productive area by 17% are planted on one ha annum^−1^.

**Figure 2 foods-12-03328-f002:**
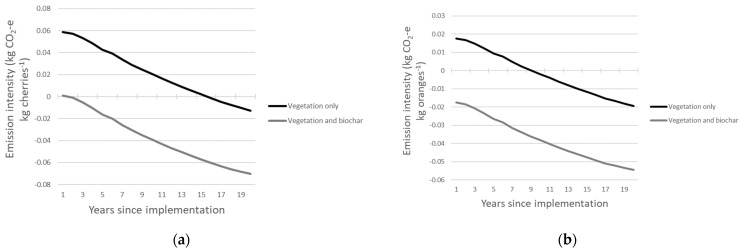
Emissions intensities (kg CO_2_-e) over a 20-year period for the production of 1 kg of (**a**) cherries cv. ‘Lapin’ and (**b**) oranges cv. ‘Washington Navel’ after the implementation of emissions reductions strategies with either C sequestration in vegetation included or C sequestered in vegetation and biochar included.

**Figure 3 foods-12-03328-f003:**
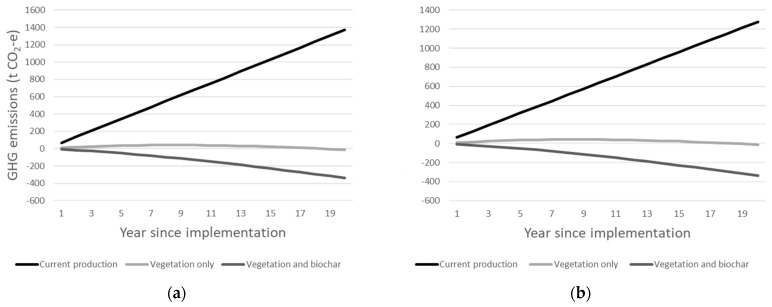
Total GHG emissions over a 20-year period for a 20 ha orchard of (**a**) cherries cv. ‘Lapin’ and (**b**) oranges cv. ‘Washington’ under current production methods, with emissions reduction strategies and C sequestered in vegetation included in the calculations, and with emissions reduction strategies and C sequestered in vegetation and biochar included in the calculations.

**Table 1 foods-12-03328-t001:** Emissions intensity (g CO_2_-e) of producing 1 kg of cherries cv. ‘Lapin’ and oranges cv. ‘Washington Navel’ under current production business as usual (BAU) and with emissions reductions strategies implemented (mitigation) and the % reduction in GHG emissions for each emissions category as calculated using life cycle assessment.

		Cherry cv. ‘Lapin’			Orange cv. ‘Washington Navel’		
	Emissions Source Category	BAU (g CO_2_-e/kg)	Mitigation (g CO_2_-e/kg)	Reduction	BAU (g CO_2_-e/kg)	Mitigation (g CO_2_-e/kg)	Reduction
**Pre-farm**	Diesel production	1.1	0.5	59%	0.8	0.4	48%
	Fertiliser production	72.4	18.9	74%	14.5	5.2	64%
	Irrigation system	0.3	0.3	0%	0.3	0.3	0%
	Transport	0.7	0.7	0%	0.4	0.4	0%
	Plant protection products	3.2	3.2	0%	5.1	5.1	0%
	Irrigation electricity	5.2	0.3	94%	85.5	0.9	99%
	**Total pre-farm**	82.8	23.8	71%	106.5	12.2	89%
**On-farm**	Fertiliser emissions	26.1	4.4	83%	5.6	3.3	42%
	Tractor operations	28.0	6.5	77%	15.5	6.0	61%
	**Total on-farm**	54.1	10.9	80%	21.1	9.3	56%

## Data Availability

The data presented in this study are available on request from the corresponding author.

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
