# Peer review of "Emissions Reduction Strategies for the Orange and Cherry Industries in New South Wales"

_foods, 2023, doi:10.3390/foods12183328_

Round 1

Reviewer 1 Report

Title

It does not clearly represent the study presented.

Abstract

What is 8?

Introduction

Carbon is abbreviated as C but spelled as full afterward.

Similar studies must have reviewed too.

Materials and Methods

The use of units is confusing L and ML is it litter and milliliter? Check also w versus W for watts.

Procedures used in “the ecoinvent v3.9 [29] or AusLCI [19] databases” may be explained for students and new researchers.

What is GWP100 and how its values for CH4 from fossil sources, CH4 from natural sources and N2O were determined as 30, 28 and 265, respectively?

Results

Transformation of data from Figure 1 to Figure 2 must be explained, in the Materials and Methods, or in the Results section.

The Results in Figure 3 must be further elaborated. The figure label and y-axis title are mixed.

Table 1 entries / data are not clear. The relevant text must explain generation of these datasets.

Conclusion

Too short.

Author Response

Reviewers’ comments

Authors response

Reviewer 1 

Title – It does not clearly represent the study presented

We have changed to title to “Emissions reduction strategies for orange and cherry production systems in New South Wales, Australia”

Abstract – What is 8

The ‘8’ has been deleted.

Introduction – carbon is abbreviated as C but spelled as full afterward

Has been changed to ‘carbon’ throughout the manuscript

Material and methods

The use of units is confusing L and ML is it liter and milliliter? Check also w versus W for watts.

ML refers to a megalitre is the preferred SI prefix for large volumes of water. All instances of ‘w’ have been changed to W

Procedures used in “the ecoinvent v3.9 [29] or AusLCI [19] databases” may be explained for students and new researchers.

We have clarified this on lines 200-204

What is GWP100 and how its values for CH4 from fossil sources, CH4 from natural sources and N2O were determined as 30, 28 and 265, respectively?

This has been explained and referenced on line 217 - 219

Results

Transformation of data from Figure 1 to Figure 2 must be explained, in the Materials and Methods, or in the Results section.

We are unclear about this comment. The authors have added additional detail in the results section on the productivity impacts of shelterbelts plantings and additional information in the materials and methods.

The Results in Figure 3 must be further elaborated. The figure label and y-axis title are mixed.

We have updated the caption for Figure 3 and have added details to the Results section.

Table 1 entries / data are not clear. The relevant text must explain generation of these datasets.

Table 1 has been updated and the data inputs are described in more detail in the materials and method section.

Conclusion

Too short.

The conclusion has been revised, lines 323-331

Reviewer 2 Report

This is an interesting paper focusing on climate change mitigation opportunities for orange and cherry production in NSW. I have these comments/suggestions with the sole aim of improving the final version of this manuscript:

From the introduction, it is important for authors to present a more robust background information especially on the the global production statistics of orange and cherry. What is the rank of Australia in the global production of these two important crops? Kindly present it. What are others important characteristics of these two crops in the Australian horticultural industries? You may need to present a a well-articulated gap(s) in knowledge for this study.

From the methods and materials, the data description and collection procedures were grosdly inadequate. Present the data used for this study in a comprehensive manner. Even if it is a secondary sourced data  I feel presenting no data and no concise data collection/source description will made it difficult for the potential readers to understand the workings of this study. Kindly do the needful.

Before the conclusion section, kindly present a sub-section with the heading "limitations of the study" and "areas for further research"

Please, present some recommendations enamating from this study after the conclusion section.

Thank you.

Fine

Author Response

Introduction

From the introduction, it is important for authors to present a more robust background information especially on the the global production statistics of orange and cherry. What is the rank of Australia in the global production of these two important crops? Kindly present it. What are others important characteristics of these two crops in the Australian horticultural industries? You may need to present a a well-articulated gap(s) in knowledge for this study.

Background information on global statistics for orange and cherry production has been added to provide context on lines 40-43 and 47-50. The purpose of the study and the knowledge gap is now stated more clearly on line 99 – 103.

Materials and methods

From the methods and materials, the data description and collection procedures were grosdly inadequate. Present the data used for this study in a comprehensive manner. Even if it is a secondary sourced data  I feel presenting no data and no concise data collection/source description will made it difficult for the potential readers to understand the workings of this study. Kindly do the needful.

More detail has been added about the input data to the materials and method section.

Two tables have been added listing data assumptions for orchard operations and now comprise supplementary materials.

Discussion

Before the conclusion section, kindly present a sub-section with the heading "limitations of the study" and "areas for further research"

The limitation and areas for further research is now included on line 322-327

Conclusion

Please, present some recommendations enamating from this study after the conclusion section.

This is now included in the revised conclusion on line 332-339

Round 2

Reviewer 1 Report

N/A

Reviewer 2 Report

The editor can make final publication decision on this revised version of the manuscript. Thank you.